# The Incidence and Predictors of Solid- and Hematological Malignancies in Patients with Giant Cell Arteritis: A Large Real-World Database Study

**DOI:** 10.3390/ijerph18147595

**Published:** 2021-07-16

**Authors:** Lior Dar, Niv Ben-Shabat, Shmuel Tiosano, Abdulla Watad, Dennis McGonagle, Doron Komaneshter, Arnon Cohen, Nicola Luigi Bragazzi, Howard Amital

**Affiliations:** 1Department of Medicine ‘B’, The Zabludowicz Center for Autoimmune Diseases, Sheba Medical Center, Ramat Gan 5265601, Israel; nivben7@gmail.com (N.B.-S.); shmil.t@gmail.com (S.T.); watad.abdulla@gmail.com (A.W.); Howard.Amital@sheba.health.gov.il (H.A.); 2Sackler Faculty of Medicine, Tel-Aviv University, Tel Aviv 6997801, Israel; 3Section of Musculoskeletal Diseases, Leeds Institute of Rheumatic and Musculoskeletal Medicine, University of Leeds and Chapel Allerton Hospital, Chapeltown Road, Leeds LS7 4SA, UK; D.G.McGonagle@leeds.ac.uk; 4Chief Physician’s Office, Clalit Health Services, Tel-Aviv 6209813, Israel; doronko1@clalit.org.il (D.K.); arcohen@clalit.org.il (A.C.); 5Siaal Research Center for Family Medicine and Primary Care, Faculty of Health Sciences, Ben-Gurion University of the Negev, Beer-Sheva 8410501, Israel; 6Laboratory for Industrial and Applied Mathematics (LIAM), York University, Toronto, ON M3J 1P3, Canada; robertobragazzi@gmail.com

**Keywords:** giant cell arteritis, temporal arteritis, vasculitis, autoinflammation, malignancy

## Abstract

Background: The association between giant cell arteritis (GCA) and malignancies had been widely investigated with studies reporting conflicting results. Therefore, in this study, we aimed to investigate this association using a large nationwide electronic database. Methods: This study was designed as a retrospective cohort study including GCA patients first diagnosed between 2002–2017 and age, sex and enrollment time-matched controls. Follow-up began at the date of first GCA-diagnosis and continued until first diagnosis of malignancy, death or end of study follow-up. Results: The study enrolled 7213 GCA patients and 32,987 age- and sex-matched controls. The mean age of GCA diagnosis was 72.3 (SD 9.9) years and 69.1% were women. During the follow-up period, 659 (9.1%) of GCA patients were diagnosed with solid malignancies and 144 (2.0%) were diagnosed with hematologic malignancies. In cox-multivariate-analysis the risk of solid- malignancies (HR = 1.12 [95%CI: 1.02–1.22]), specifically renal neoplasms (HR = 1.60 [95%CI: 1.15–2.23]) and sarcomas (HR = 2.14 [95%CI: 1.41–3.24]), and the risk of hematologic malignancies (HR = 2.02 [95%CI: 1.66–2.47]), specifically acute leukemias (HR = 1.81 [95%CI: 1.06–3.07]), chronic leukemias (HR = 1.82 [95%CI: 1.19–2.77]), Hodgkin’s lymphomas (HR = 2.42 [95%CI: 1.12–5.20]), non-Hodgkin’s-lymphomas (HR = 1.66: [95%CI 1.21–2.29]) and multiple myeloma(HR = 2.40 [95%CI: 1.63–3.53]) were significantly increased in GCA patients compared to controls. Older age at GCA-diagnosis (HR = 1.36 [95%CI: 1.25–1.47]), male-gender (HR = 1.46 [95%CI: 1.24–1.72]), smoking (HR = 1.25 [95%CI: 1.04–1.51]) and medium-high socioeconomic status (HR = 1.27 [95%CI: 1.07–1.50]) were independently associated with solid malignancy while age (HR = 1.47 [95%CI: 1.22–1.77]) and male-gender (HR = 1.61 [95%CI: 1.14–2.29]) alone were independently associated with hematologic- malignancies. Conclusion: our study demonstrated higher incidence of hematologic and solid malignancies in GCA patients. Specifically, leukemia, lymphoma, multiple myeloma, kidney malignancies, and sarcomas. Age and male gender were independent risk factors for hematological malignancies among GCA patients, while for solid malignancies, smoking and SES were risk factors as well.

## 1. Introduction

Giant cell arteritis (GCA) is an inflammatory disorder of large and medium-sized arteries that preferentially involves branches of the external carotid artery [1]. Epidemiologic studies demonstrated disease predominance among elderly females and individuals of Northern European ancestry. The etiopathogenesis of GCA is uncertain, with evidence suggesting roles played by certain genetic backgrounds as well as immunosenescence as driving forces in the disease pathogenesis [2,3]. The association between chronic inflammation and tumorigenesis had been widely investigated. Over the years, various shared mechanisms were postulated to be at the core of both phenomena such as excessive cell replication, resistance to growth inhibition, and enhanced angiogenesis [4,5].

Several rheumatologic diseases were associated with increased risk of malignancy. The association between Sjogren’s syndrome, rheumatoid arthritis (RA) and systemic sclerosis with malignancy had been widely established by previous studies. RA and Sjogren’s syndrome were found to be associated particularly with lymphomas, whereas systemic sclerosis patients were associated with lung cancer [6,7,8,9,10]. The results between systemic lupus erythematosus (SLE) and malignancy were less equivocal [11,12,13]. Regarding vasculitides, polyarteritis nodosa, and antineutrophil cytoplasmic antibody (ANCA) associated vasculitis were shown to have a positive association with malignancies, predominantly of hematologic origin [14,15,16].

Data regarding the association between GCA and the occurrence of malignancies has furnished conflicting results. Several studies had found no significant association [17,18,19,20], while others had shown positive association between GCA and the incidence of malignancy [21,22,23], including a recent meta-analysis [24]. The above-mentioned meta-analysis demonstrated a 14% excess risk of malignancy compared with non-GCA/PMR participants. Among the few studies that addressed specific types of cancers, higher rates of solid tumors (melanoma, stomach, lung, prostate, kidney, nervous system, and endocrine malignancies) as well as non-Hodgkin’s lymphoma (NHL), multiple myeloma, and leukemia among GCA patients were reported [23,25]. While breast, endometrial, and gastrointestinal tract malignancies were reported to be at significantly lower rates in these patients [23,25].

Dysphagia, anemia, and an abnormal temporal artery on palpation at the time of GCA diagnosis were found to be associated with the incidence of malignancy [20]. However, most of these studies had relatively small cohorts. Among the two studies with large cohorts, the first did not directly address this issue and did not investigate different types of cancer [26], while the other lacked a comparison cohort and used the general population cancer rates as a reference [23]. Furthermore, no study so far had investigated the potential effect of socioeconomic status (SES), obesity and smoking, which are known to be related to malignancies and may act as potential confounders.

Therefore, in this study, we investigated the relation between GCA and malignancy in a large population-based cohort, considering, for the first time, potential demographic parameters in a multivariate model.

## 2. Materials and Methods

### 2.1. Data Source

This study was approved by the CHS Ethics Committee in Tel Aviv, Israel. Approval number 0212-17-COM. No informed consent was needed (existing database). For this study, we used Clalit Healthcare Services (CHS) electronic database. CHS is the largest health maintenance organization (HMO) in Israel and serves approximately 4.5 million members, which comprises over 50% of the Israel population. Data in CHS database is collected continuously from pharmaceutical, medical, and administrative operating systems, and is going through a process of diagnosis validation by logistic checks. The CHS database was shown to have high validity in previous studies, including those involving diagnoses from rheumatology [27,28,29,30] and oncology [31] disciplines. Moreover, the current cohort of GCA patients was used by our group in a recent study addressing mortality in GCA [32].

### 2.2. Study Population and Design

This study was designed as a retrospective cohort study comparing incident GCA cases to age- and sex-matched controls. Our GCA cohort included all patients with at least one documented diagnosis of GCA (ICD9 code 446.5) made in primary care centers, inpatients and outpatient clinics, or hospitalization discharge letters, between 1 January 2002 to 31 December 2017. Patients that were under the age of 50 years at the time of the first diagnosis and patients who had a recorded diagnosis of GCA before 1 January 2002 were excluded. Controls were age, gender, and enrollment date-matched and included patients without a diagnosis of GCA that were randomly assigned from the CHS electronic database in a ratio of about 5:1. Follow-up began at the date of first GCA diagnosis and continued until the earliest of the following: occurrence of malignancy, death or end of study follow-up on 1 September 2018.

### 2.3. Study Variables

The definition of a malignancy was based on a documented diagnosis of that type of malignancy in the medical records, as registered in the CHS database. Based on date of first diagnosis, malignancies were categorized as occurring before GCA diagnosis (enrollment time for controls) or after. Malignancies that were diagnosed before the occurrence of GCA (including those diagnosed before 2002), were addressed as baseline malignancies, while those that were diagnosed during follow-up (after the occurrence of GCA) were addressed as incident malignancies.

For each subject, age, gender and socioeconomic status (SES) at enrollment time were obtained. SES was defined according to a poverty index which is based on the member’s residence area. The poverty index was defined during the 2008 National Census, and considered average household income, education, crowding, and car ownership. We divided the population into three categories based on terciles. Obesity was considered as having a Body-Mass-Index ≥ 30 kg/m^2^ in a measure during the year of enrollment. Smoking was dichotomized into ever vs. never smoked at the time of enrollment. All data were obtained from the CHS electronic database, which was previously demonstrated to have 90 to 100% degree of accuracy [27].

### 2.4. Statistical Analysis

Differences in baseline characteristics between different groups of independent variables were compared using t-test or Mann–Whitney U test for continuous variables, and χ^2^ test for categorical variables. Rates of malignancies were compared between GCA patients and controls in three time periods: at any time during subject life period, before the diagnosis of GCA/enrollment and after the diagnosis of GCA/enrollment. Survival analysis demonstrating cancer-free cumulative frequency was performed using Kaplan–Meier method with a post hoc log-rank comparison of GCA patients and controls. Cox proportional hazard model was used for univariate and multivariate survival analyses. Malignancy was considered as an event, and hazard ratios (HR) were calculated comparing GCA and controls. This analysis included only patients without diagnosis of malignancy before enrollment. The multivariate model accounted for age, gender, smoking, SES, and obesity. Patients with a diagnosis of any malignancy before enrollment were excluded from survival analyses. Statistical analysis was performed using the commercial software “Statistical Package for the Social Sciences” (SPSS for Windows, V.23.0 (IBM SPSS Statistics, Armonk, NY, USA).

## 3. Results

### 3.1. Cohort Characteristics

The study population included 7213 GCA patients and 32,987 age- and sex-matched controls. The mean age of GCA diagnosis was 72.3 ± 9.9 years (median 73.1 years) and 69.1% were women. No statistically significant difference was found in SES between groups. The GCA patients had significantly higher rates of smoking (24.8% vs. 19.8%, *p* < 0.001) and obesity (24.2% vs. 16.3%, *p* < 0.001) than controls (Table 1).

### 3.2. Cancer Rates in GCA Patients

Baseline rates of hematologic cancers (1.9% vs. 1.2%, *p* < 0.001) specifically, multiple myeloma (0.4% vs. 0.2%, *p* < 0.001), non-Hodgkin’s lymphoma (0.9% vs. 0.6%, *p* < 0.001), and Hodgkin’s lymphoma (0.3% vs. 0.1%, <0.005), as well as prostate cancer (5.5% vs. 4.4%, *p* < 0.05), were significantly higher in GCA patients compared to controls. During follow-up 659 (9.1%) of GCA patients were diagnosed with solid cancer and 144 (2.0%) were diagnosed with hematologic cancer. Regarding cancers first diagnosed after enrollment (incident cases), rates of solid cancers (9.1% vs. 8.4%, *p* < 0.05), specifically kidney (0.7% vs. 0.4%; *p* < 0.05) and sarcoma (0.4% vs. 0.2%; *p* < 0.001) and rates of hematologic cancers (2% vs. 1%, *p* < 0.001), specifically acute-leukemia (0.3% vs. 0.1%; *p* < 0.05), chronic leukemia (0.4% vs. 0.2%; *p* < 0.01), multiple myeloma (0.5% vs. 0.2%; *p* < 0.001), non-Hodgkin’s lymphoma (0.7% vs. 0.4%; *p* < 0.005), and Hodgkin’s lymphoma (0.2% vs. 0.1%; *p* < 0.05) were higher in GCA patients compared to controls (Table 2). Rates of CNS, bone, larynx, and pancreas malignancies were also examined and were comparable for GCA and controls (data not shown).

### 3.3. Cancer Risk in GCA Patients

At the univariate cox model, GCA patients significantly demonstrated increased risk of solid cancers in general (HR 1.13 [95%CI 1.03–1.23]), specifically lung cancer (HR 1.30 [95%CI 1.03–1.64]), kidney cancer (HR 1.62 [95%CI 1.16–2.25]) and sarcomas (HR 2.10 [95%CI 1.38–3.17]), as well as increased risk of overall hematological malignancies (HR 1.62 [95%CI 1.16–2.25]) and more specifically acute leukemia (HR 1.94 [95%CI 1.15–3.26]), chronic leukemia (HR 1.94 [95%CI 1.29–1.92]), Hodgkin’s lymphoma (HR 2.75 [95%CI 1.31–5.78]), non-Hodgkin’s lymphoma (HR 1.73 [95%CI 1.26–2.38]) and multiple myeloma (HR 2.44 [95%CI 1.67–3.59) compared to controls (Table 3). At the cox multivariate model, adjusting for age, sex, SES, obesity and smoking status, GCA patients demonstrated significantly increased risk of solid cancers in general (HR 1.12 [95%CI 1.02–1.22]), specifically renal cancer (HR 1.60 [95%CI 1.15–2.23]) and sarcomas (HR 2.14 [95%CI 1.41–3.24]), as well as increased risk of hematologic cancers in general (HR 2.02 [95%CI 1.66–2.47]), specifically acute-leukemia (HR 1.81 [95%CI 1.06–3.07]), chronic-leukemia (HR 1.82 [95%CI 1.19–2.77]), Hodgkin’s lymphoma (HR 2.42 [95%CI 1.12–5.20]), non-Hodgkin’s lymphoma (HR 1.66 [95%CI 1.21–2.29]) and multiple myeloma (HR 2.40 [95%CI 1.63–3.53]). The risk of gastric cancer (HR 0.61 [95%CI 0.37–0.98]) was significantly lower in GCA patients compared to controls (Table 3).

Kaplan–Meier survival curves, assessing cumulative cancer-free survival (Figure 1) demonstrate survival curves for GCA patients in both solid (Figure 1A) and hematologic (Figure 1B) cancers. The average time (mean [months] ± SD) to the diagnosis of any malignancy was significantly shorter in GCA patients (48.6 ± 41.3) compared to controls (58.1 ± 43.6; *p* < 0.001).

### 3.4. Predictors of Cancer in GCA Patients

Older age at diagnosis of GCA (HR 1.36 [95%CI 1.25–1.47], for every 10 years), male gender (HR 1.46 [95%CI 1.24–1.72]), smoking (HR 1.25 [95%CI 1.04–1.51]) and medium-high SES (HR 1.27 [95%CI 1.07–1.50]) were independent risk factors for solid cancers in a multivariate analysis. For hematological cancers, only older age at diagnosis (HR 1.66 [95%CI 1.21–2.29]) and male gender (HR 1.66 [95%CI 1.21–2.29]) were significant predictors (Table 4).

## 4. Discussion

In this large nationwide population-based study, we found an increased risk for sarcomas, kidney malignancies, and overall solid malignancies in GCA patients compared to controls. GCA was also separately associated with an increased risk of all hematologic malignancies, (leukemia, lymphoma, and multiple myeloma).

The GCA cohort in our study is consistent in terms of age of diagnosis and female to male ratio with previous reports [33]. Malignancy rates were also appropriate for age [34]. The increased risk for hematologic, sarcoma, and kidney malignancies seen in our study corresponds with several large scale studies [22,23,24]. Most notably is the study of Sundquist et al. [23], which encompassed 36,000 GCA cases with concurrent polymyalgia rheumatica from a Swedish registry. Similar to our results, Sundquist et al. [23] reported increased risk of hematologic malignancies with a standardized incidence ratio (SIR) ranging from 1.32 to 2.69, renal malignancies with a SIR of 1.56, and sarcomas with a SIR of 4.92 (mainly in the first year after GCA diagnosis). Higher rates of lung malignancies were also reported in the Swedish report, which is comparable with our results of the unadjusted model; however, after adjusting these results to smoking, which was more common in GCA patients, these results were no longer statistically significant. Unlike our study, Sundquist et al. [23] did not use a matched comparison cohort and included only hospitalized patients. Another interesting finding of our study is the decreased risk for gastric cancer seen in GCA patients. This association was previously reported by Stamatis et al. [25], which hypothesized that obesity could act as a confounder in this case due to its known association with gastric cancer. However, this association was demonstrated in our study after an adjustment to obesity. Therefore, it is unlikely to be the cause. Further inquiry is required in order to clarify this untrivial association.

In our GCA cohort older age at diagnosis and male gender were predictors of both solid and hematological malignancies, while smoking and medium-high socioeconomic status were significant predictors only for solid malignancies. These results are rather consistent with risk factors for malignancy in the general population, and we were the first to exhibit them in GCA patients [35,36].

There is no well-formulated theory explaining this linkage tying an increased rate of malignancies with coexisting GCA. Various reports had highlighted the role of ongoing inflammation as a driving force in the pathogenesis and progression of malignancies. For example, Craver et al. [37] investigated the role of inflammation in myeloid line dysregulation and malignancies, and noted that chronic inflammation may promote hematopoietic stem cell exhaustion, promoting the emergence of mutant clones thus promoting the development of myeloid malignancy [37]. Moreover, they reported that higher concentrations of serum inflammatory cytokines in myeloid malignancies, which were shown to be associated with disease initiation, burden, and progression, as well as worsened survival outcome [37].

Smedby et al. [38] conducted a large population-based case-controlled study that demonstrated the increased risk of NHL in patients with certain autoimmune/inflammatory disease (RA, Sjogren’s syndrome, SLE and celiac disease), and Baecklund et al. [39] postulated that the chronic B-cell stimulation and antigenic drive play a role in inflammation-related lymphogenesis. Moreover, systemic autoimmune features, such as increased resistance to apoptosis, were reported to further enhance the carcinogenic effects of B cell proliferation [40]. As for kidney cancer, an association between malignancy and antineutrophil cytoplasmic antibody (ANCA)-associated vasculitis had been previously demonstrated, and in particular, renal carcinoma was found to be associated with granulomatosis with polyangiitis (Wegener’s granulomatosis), suggesting a carcinogenic effect of the disease process itself and chronic stimulation of the immune system and perhaps the result of exposure to alkylating agents [14,15]. There might be a common pathogenetic pathway that leads to both GCA and kidney cancer, probably via the involvement of the renal vasculature.

Another possible explanation for the increased risk of hematologic malignancies and GCA can be a shared trigger for both conditions. Several studies have suggested a role for environmental factors, such as viral genome integration by oncogenic viruses (EBV, HTLV-1, HHV-8), which has been associated with the pathogenesis of specific NHL subtypes [41]. Moreover, various microbe and viral sequences, including varicella zoster virus, have been detected in temporal artery lesions [42]. There may be a shared trigger for GCA and hematologic malignancies that has yet to be discovered.

A setting of close temporal association with the onset of GCA may represent a paraneoplastic syndrome triggered by the anti-tumor immune response [43]. This mechanism is reinforced by a study demonstrated a correlation between vasculitis disease activity to the onset, resolution, and recurrence of solid tumors [44].

Our investigation has several strengths; the population-based design, which is based on big data analysis of real-life populations. Moreover, the large database ensures the inclusion and representation of the whole population, thereby facilitating generalization and avoiding referral bias. However, various limitations warrant consideration. First, we had no data regarding temporal-artery biopsies, which is the gold-standard of GCA diagnosis. Thus, we were unable to distinguish those diagnosed based on pathology from those diagnosed based on laboratory, imaging, and clinical findings. However, it is worth noting that that temporal biopsy is not mandatory for the diagnosis of GCA, which can be clinically done with the presence of three out of five criteria. Furthermore, previous studies showed the same features and outcomes in biopsy-proven GCA patients compared to those diagnosed clinically [20,24,25]. Regarding the ascertainment of cancer diagnosis and co-morbidities, our cohort had similar rates to those reported by the Israeli ministry of health [34] and smoking and obesity rates matched those of the Israeli population of the same age [45,46]. In addition, our data were limited regarding the use of systemic corticosteroids and other immunosuppressants commonly used for the treatment of GCA patients. Corticosteroids is the cornerstone in the treatment of GCA, and all patients with GCA are advised to start with corticosteroids treatment [47]. In earlier studies, corticosteroids therapy was linked to NHL, yet more recent ones did not confirm this observation NHL [11,38,48]. Moreover, our study investigated the incidence of malignancy in GCA patients, and did not investigate malignancy as a cause of death in GCA patients, as was sown in a previous large recent study [49] to be lower in GCA patients. The latter result of decreased risk of death due to cancer in GCA patients can be explained by an increased cardiovascular death risk. Finally, detection bias cannot be ruled out, as GCA diagnosed patients may be exposed to higher rates of physician and imaging examinations, which may enhance the probability of cancer diagnosis. In addition, we do not have data regarding the use of cyclophosphamide or azathioprine for refractory cases, but if such cases occurred, they are probably extremely rare with negligible effect.

In conclusion, our study demonstrated higher incidence of hematologic and solid malignancies in GCA patients. Specifically, leukemia, lymphoma, multiple myeloma, kidney malignancies and sarcomas. Age and male gender were independent risk factors for hematological malignancies among GCA patients, while for solid malignancies, smoking and SES were risk factors as well.

## Figures and Tables

**Figure 1 ijerph-18-07595-f001:**
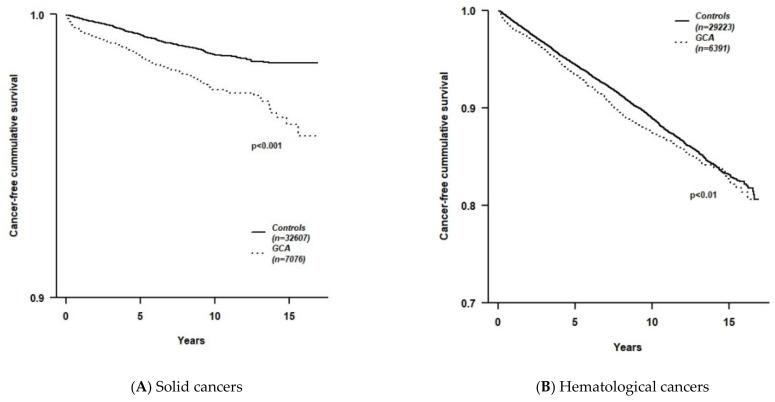
Kaplan–Meier survival curves comparing cancer-free survival time between GCA patients and age- and sex-matched controls. (**A**) Solid cancers; (**B**) Hematological cancers.

**Table 1 ijerph-18-07595-t001:** Basic characteristics of the study population.

Characteristic	GCA (*n* = 7213)	Controls (*n* = 32,987)	*p*-Value
Age, mean ± SD; median	72.3 ± 9.9; 73.1	72.1 ± 9.8; 73.1	0.16
Gender (females), n (%)	4987 (69.1%)	22,929 (69.5%)	0.53
Follow-up duration (years) mean ± SD; median	7.1 ± 4.4; 6.5	7.4 ± 4.4; 6.9	<0.001
SES, n (%) ^a^			0.72
Low	2368 (33.1%)	11,091 (33.0%)	
Medium	3049 (42.6%)	14,103 (43.1%)	
High	1737 (24.3%)	7828 (23.9%)	
Smoking, n (%)	1790 (24.8%)	7005 (21.2%)	<0.001
Obesity, n (%)	1747 (24.2%)	6949 (21.1%)	<0.001
Death, n (%)	2425 (33.6%)	10,481 (31.8%)	<0.01

GCA, giant cell arteritis; SD, standard-deviation; SES, socioeconomic status. ^a^ Available for 99.2% of data.

**Table 2 ijerph-18-07595-t002:** Malignancy rates by site, relative to the time of GCA diagnosis, or enrollment time for controls.

Cancer Site	GCA (*n* = 7213) n (%)	Controls (*n* = 32,987) n (%)	*p*-Value
Solid cancers	1378 (19.1%)	6115 (18.5%)	0.26
Before diagnosis/enrollment	798 (11.1%)	3634 (11%)	0.91
After diagnosis/enrollment	659 (9.1%)	2765 (8.4%)	<0.05
Oropharyngeal	47 (0.7%)	221 (0.7%)	0.93
Before diagnosis/enrollment	7 (0.1%)	25 (0.1%)	0.49
After diagnosis/enrollment	40 (0.6%)	196 (0.6%)	0.73
Thyroid	63 (0.9%)	233 (0.7%)	0.14
Before diagnosis/enrollment	49 (0.7%)	177 (0.5%)	0.14
After diagnosis/enrollment	14 (0.2%)	56 (0.2%)	0.64
Lung	115 (1.6%)	484 (1.5%)	0.42
Before diagnosis/enrollment	24 (0.3%)	152 (0.5%)	0.17
After diagnosis/enrollment	91 (1.3%)	332 (1.0%)	0.054
Esophagus	13 (0.2%)	43 (0.1%)	0.29
Before diagnosis/enrollment	2 (0%)	15 (0%)	0.75
After diagnosis/enrollment	11 (0.2%)	28 (0.1%)	0.09
Stomach	40 (0.6%)	238 (0.7%)	0.14
Before diagnosis/enrollment	20 (0.3%)	92 (0.3%)	0.99
After diagnosis/enrollment	20 (0.3%)	146 (0.4%)	0.058
Colorectal	286 (4.1%)	1368 (4.0%)	0.49
Before diagnosis/enrollment	151 (2.1%)	802 (2.4%)	0.10
After diagnosis/enrollment	135 (1.9%)	566 (1.7%)	0.37
Liver and bile ducts	33 (0.5%)	132 (0.4%)	0.48
Before diagnosis/enrollment	6 (0.1%)	17 (0.1%)	0.28
After diagnosis/enrollment	27 (0.4%)	115 (0.3%)	0.75
Kidney	79 (1.1%)	306 (0.9%)	0.18
Before diagnosis/enrollment	32 (0.4%)	167 (0.5%)	0.57
After diagnosis/enrollment	47 (0.7%)	139 (0.4%)	<0.05
Bladder	134 (1.9%)	554 (1.7%)	0.29
Before diagnosis/enrollment	66 (0.9%)	281 (0.9%)	0.58
After diagnosis/enrollment	68 (0.9%)	273 (0.8%)	0.32
Breast ^a^	365 (7.3%)	1810 (7.9%)	0.18
Before diagnosis/enrollment	260 (5.2%)	1273 (5.6%)	0.35
After diagnosis/enrollment	105 (2.1%)	537 (2.3%)	0.35
Uterus ^a^	56 (1.1%)	313 (1.4%)	0.19
Before diagnosis/enrollment	31 (0.6%)	175 (0.8%)	0.32
After diagnosis/enrollment	25 (0.5%)	138 (0.6%)	0.47
Cervix ^a^	14 (0.3%)	80 (0.3%)	0.49
Before diagnosis/enrollment	11 (0.2%)	49 (0.2%)	0.87
After diagnosis/enrollment	3 (0.1%)	31 (0.1%)	0.26
Ovary ^a^	32 (0.6%)	158 (0.7%)	0.77
Before diagnosis/enrollment	12 (0.2%)	70 (0.3%)	0.55
After diagnosis/enrollment	20 (0.4%)	88 (0.4%)	0.81
Prostate ^b^	177 (8.0%)	695 (6.9%)	0.09
Before diagnosis/enrollment	122 (5.5%)	445 (4.4%)	<0.05
After diagnosis/enrollment	55 (2.5%)	250 (2.5%)	0.99
Sarcoma	58 (0.8%)	170 (0.5%)	<0.01
Before diagnosis/enrollment	26 (0.4%)	97 (0.3%)	0.35
After diagnosis/enrollment	32 (0.4%)	73 (0.2%)	<0.001
Melanoma	101 (1.4%)	433 (1.3%)	0.57
Before diagnosis/enrollment	59 (0.8%)	251 (0.8%)	0.61
After diagnosis/enrollment	42 (0.6%)	182 (0.6%)	0.73
Hematologic cancers	274 (3.8%)	704 (2.1%)	<0.001
Before diagnosis/enrollment	137 (1.9%)	380 (1.2%)	<0.001
After diagnosis/enrollment	144 (2.0%)	345 (1.0%)	<0.001
Acute Leukemia	36 (0.5%)	96 (0.3%)	<0.005
Before diagnosis/enrollment	16 (0.2%)	47 (0.1%)	0.14
After diagnosis/enrollment	20 (0.3%)	49 (0.1%)	<0.05
Chronic Leukemia	55 (0.8%)	153 (0.5%)	<0.01
Before diagnosis/enrollment	21 (0.3%)	74 (0.2%)	0.29
After diagnosis/enrollment	32 (0.4%)	79 (0.2%)	<0.01
Myelodysplastic syndrome	26 (0.4%)	68 (0.2%)	<0.05
Before diagnosis/enrollment	14 (0.2%)	37 (0.1%)	0.10
After diagnosis/enrollment	12 (0.2%)	31 (0.1%)	0.11
Multiple Myeloma	71 (1%)	135 (0.4%)	<0.001
Before diagnosis/enrollment	29 (0.4%)	59 (0.2%)	<0.001
After diagnosis/enrollment	39 (0.5%)	76 (0.2%)	<0.001
Non-Hodgkin Lymphoma	121 (1.7%)	325 (1%)	<0.001
Before diagnosis/enrollment	68 (0.9%)	182 (0.6%)	<0.001
After diagnosis/enrollment	52 (0.7%)	143 (0.4%)	<0.005
Hodgkin Lymphoma	32 (0.4%)	64 (0.2%)	<0.001
Before diagnosis/enrollment	21 (0.3%)	45 (0.1%)	<0.005
After diagnosis/enrollment	11 (0.2%)	19 (0.1%)	<0.05

Abbreviations: GCA, giant cell arteritis; ^a^ for female patients ^b^ for male patients.

**Table 3 ijerph-18-07595-t003:** Hazard ratios for the incidence of cancer in GCA patients compared to age- and sex-matched controls.

Cancer Site	Time (Months) to Malignancy ^a^ Mean (SD)	Crude HR		Adjusted ^b^ HR	
		HR	95%CI	HR	95%CI
Solid cancers ^*^	50.8 (40)	1.13	1.03, 1.23	1.12	1.02, 1.22
Oropharyngeal	52.1 (33)	0.88	0.69, 1.37	0.96	0.69, 1.35
Thyroid	44.7 (41)	1.19	0.66, 2.14	1.20	0.66, 2.16
Lung	49.1 (39)	1.30	1.03, 1.64	1.26	0.99, 1.59
Esophagus	47.9 (31)	1.86	0.93, 3.74	1.86	0.93, 3.75
Stomach^*^	51.4 (43)	0.65	0.41, 1.04	0.61	0.37, 0.98
Colorectal	43.9 (35)	1.13	0.94, 1.37	1.12	0.93, 1.36
Liver and bile ducts	51.3 (39)	1.12	0.73, 1.69	1.06	0.69, 1.62
Kidney ^**^	45.4 (40)	1.62	1.16, 2.25	1.60	1.15, 2.23
Bladder	63.4 (42.7)	1.18	0.91, 1.54	1.15	0.88, 1.50
Breast ^c^	58 (42)	0.93	0.75, 1.14	0.93	0.75, 1.14
Uterus	73.9 (47)	0.86	0.56, 1.31	0.85	0.55, 1.30
Cervix	45.5 (62)	0.46	0.14, 1.50	0.45	0.37, 1.46
Ovary ^c^	63.2 (57)	1.08	0.66, 1.76	0.96	0.58, 1.59
Prostate ^d^	50.3 (38)	1.07	0.79, 1.43	1.07	0.80, 1.43
Sarcoma ^**^	46.5 (43)	2.10	1.38, 3.17	2.14	1.41, 3.24
Melanoma	56 (39)	1.10	0.78, 1.53	1.11	0.80, 1.55
Hematologic cancers ^**^	49.4 (46)	2.02	1.66, 2.47	1.95	1.59, 2.38
Acute Leukemia ^*^	55 (56)	1.94	1.15, 3.26	1.81	1.06, 3.07
Chronic Leukemia ^**^	71.3 (51)	1.94	1.29, 1.92	1.82	1.19, 2.77
Myelodysplastic syndrome	36.3 (26)	1.83	0.94, 3.55	1.84	0.94, 3.60
Hodgkin Lymphoma ^*^	40 (43)	2.75	1.31, 5.78	2.42	1.12, 5.20
Non-Hodgkin Lymphoma ^**^	49.1 (41)	1.73	1.26, 2.38	1.66	1.21, 2.29
Multiple Myeloma ^**^	41.2 (44)	2.44	1.67, 3.59	2.40	1.63, 3.53

Abbreviations: CI, confidence interval; HR, hazard ratio; SD, standard deviation. ^a^ Among GCA patients with no prior malignancy; ^b^ The model included the variables: age, gender, socioeconomic status, smoking, and obesity; ^c^ Only females were included in the model; ^d^ Only males were included in the model. ^*^ Statistical significance at 95% confidence level. ^**^ Statistical significance at 99% confidence level.

**Table 4 ijerph-18-07595-t004:** Independent predictors for solid and hematologic malignancies in GCA cohort.

	HR (95%CI) ^a^	
	Solid Cancers	Hematologic Cancers
Age at diagnosis (every 10 years increment)	1.36 (1.25–1.47) ^**^	1.47 (1.22–1.77) ^**^
Gender (males vs. females)	1.46 (1.24–1.72) ^**^	1.61 (1.14–2.29) ^**^
Smoking (ever vs. never)	1.25 (1.04–1.51) ^*^	1.27 (0.85–1.90)
Obesity (BMI > 30 kg/m^2^ vs. normal)	1.14 (0.94–1.38)	1.14 (0.75–1.73)
SES (high and medium vs. low)	1.27 (1.07–1.50) ^**^	1.25 (0.86–1.82)

Abbreviations: CI, confidence interval; HR, hazard ratio; BMI, body mass index; SES, socioeconomic status. ^a^ The model included the variables: age, gender, socioeconomic status, smoking, and obesity. ^*^ Statistical significance at 95% confidence level. ^**^ Statistical significance at 99% confidence level.

## Data Availability

Not applicable.

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
