# Peer review of "The Incidence and Predictors of Solid- and Hematological Malignancies in Patients with Giant Cell Arteritis: A Large Real-World Database Study"

_ijerph, 2021, doi:10.3390/ijerph18147595_

Round 1

Reviewer 1 Report

Ms. Ref. No.: ijerph-1269786

Title: The rate, risk and predictors of solid- and hematological malignancies in patients with giant cell arteritis: A large real-world database study

Overview

Elevated cancer risk in patients with rheumatologic diseases is of major concern and is driven not only by disease- but also by treatment-related factors. Therefore, this study is on a topic of interest in the field of giant cell arteritis (GCA). The authors investigate cancer risk in a population of GCA patients and describe an increased risk for hematologic and solid cancer in these patients. They identify male gender and older age to be associated with cancer risk. Strengths of this study are thorough data presentation, the large cohort, the comparator group and the adjustment for confounders such as smoking status, age or obesity. Limitations are related to the retrospective study design and include missing details about GCA disease characteristics such as diagnostic and treatment parameters.

Minor comments  

Methods:

Cancers before GCA diagnosis/ enrollment: Were only cancers that were diagnosed in the study period (2002-2017) captured or did you include cancers diagnosed before 2002? Please consider to include this information in the methods part.

2.1

Consider to rephrase the sentence: CHS database was used in several studies before and was demonstrated to have a high validity of diagnoses for chronic diseases including rheumatological field [27–30] oncological [31].

Results

3.1

Please include the follow-up duration for both groups.

3.2

You might consider to start the paragraph with the baseline rates and continue with the follow-up numbers.

Table 2: You might consider to rephrase “out of male/female” e.g. for male/female patients.

3.3

Titel: Cancer risk in GCA patients. You might consider to include the term “after GCA diagnosis/ enrollment”

Sentence: At the cox multivariate model, adjusting for age, sex, SES and cancer risk factors, GCA patients demonstrated significantly increased risk of solid cancers in general (HR 1.12… You might consider to change the term “cancer risk factors” to “obesity and smoking status” to be more precise.

Figure 1: Please include numbers below the figure (at least numbers at risk).

Discussion

You might consider to comment on the fact that a large number of patients were diagnosed with cancer before GCA diagnosis/ enrollment (in fact even more than after GCA diagnosis /enrollment). Patients with malignancy are at risk to develop a second malignancy. Only for hematological cancers, the numbers were significantly higher in GCA patients compared to controls before GCA diagnosis/ enrollment.

Paragraph 5 and 6: Last sentence is a bit too speculative. Please consider to rephrase. (There might be a common pathogenetic pathway that leads both GCA and kidney cancer… . There may be a shared trigger for GCA and hematologic malignancies that has yet to be discovered… .)

Nice work!

Reviewer 2 Report

Giant cell arteritis is an inflammatory disorder of blood vessels that exclusively occurs in patients older than 50 years with a higher prevalence in women. Although the previous meta-analysis studies reported a significantly increased malignancy risk for patients with GCA than controls, individual studies still vary with conflicting conclusions. In this manuscript, the authors performed a large scale retrospective-cohort and follow-up study to elucidate the malignancies incidence with GCA patients. They found a higher incidence of hematologic and malignancies among GCA patients. Of note, this study also includes new confounder factors, like socioeconomic status (SES), obesity, and smoking. Overall, this study is well designed and presented. I have few suggestions:

  1. The follow-up time should be provided in the study design.

  1. Add the symbol of Statistical significance in Table 1 and 2.

  1. One of the important studies related to this topic should be cited and discussed. (10.1186/s13075-019-1945-4)

Reviewer 3 Report

The study by Dar et al is a large important study reporting on solid and hematological malignancies in the context of GCA. The authors show the risk is increased in GCA patients, especially so for those with obesity or those who smoke. It's a well performed rétrospective study which unfortunately lacks data on treatment of GCA especially corticosteroids but the authors address this limitation in the discussion. I think this study is ready for publication and I just have one question: is tocilizumab used often enough in Israel to influence the current data?

Author Response

The trend to start Tocilizumab in the early stages of the disease, according to the International guidelines, had gain momentum only in recent years in Israel. We believe, its use in the period of the study follow-up was negligible.